# Serologic and Molecular Diagnosis of *Anaplasma platys* and *Ehrlichia canis* Infection in Dogs in an Endemic Region

**DOI:** 10.3390/pathogens9060488

**Published:** 2020-06-19

**Authors:** Bianca Lara, Anne Conan, Mary Anna Thrall, Jennifer K. Ketzis, Gillian Carmichael Branford, Sreekumari Rajeev

**Affiliations:** Department of Biomedical Sciences, Ross University School of Veterinary Medicine, Saint Kitts, KN 0101, West Indies; biancalara@students.rossu.edu (B.L.); aconan@hotmail.fr (A.C.); MThrall@rossvet.edu.kn (M.A.T.); JKetzis@rossu.edu (J.K.K.); GCarmichaelBranford@rossu.edu (G.C.B.)

**Keywords:** *Anaplasma platys*, *Ehrlichia canis*, diagnosis, point-of-care test, PCR

## Abstract

*Anaplasma platys* and *Ehrlichia canis* are obligate intracellular, tick-borne rickettsial pathogens of dogs that may cause life-threatening diseases. In this study, we assessed the usefulness of PCR and a widely used commercial antibody-based point-of-care (POC) test to diagnose *A. platys* and *E. canis* infection and updated the prevalence of these pathogens in dogs inhabiting the Caribbean island of Saint Kitts. We detected *A. platys* in 62/227 (27%), *E. canis* in 84/227 (37%), and the presence of both in 43/227 (19%) of the dogs using PCR. POC testing was positive for *A. platys* in 53/187 (28%), *E. canis* in 112/187 (60%), and for both in 42/187 (22%) of the samples tested. There was only a slight agreement between *A. platys* PCR and POC test results and a fair agreement for *E. canis* PCR and POC test results. Our study suggests that PCR testing may be particularly useful in the early stage of infection when antibody levels are low or undetectable, whereas, POC test is useful when false-negative PCR results occur due to low bacteremia. A combination of PCR and POC tests may increase the ability to diagnose *A. platys* and *E. canis* infection and consequently will improve patient management.

## 1. Introduction

*Anaplasma platys* and *Ehrlichia canis* are Gram-negative obligate intracellular, tick-borne rickettsial pathogens infecting dogs. *A. platys* infects canine platelets and is the causative agent of infectious canine cyclic thrombocytopenia (ICCT) [1,2]. The infection is often mild or asymptomatic but can become fatal due to severe thrombocytopenia and subsequent potential for hemorrhaging [3,4]. *E. canis* infection causes canine monocytic ehrlichiosis (CME) and can present as an acute, subclinical, or chronic debilitating disease, often manifesting with a wide range of nonspecific clinical signs [3,5]. Chronic infections can go unnoticed until severe laboratory findings such as pancytopenia, hyperglobulinemia, and proteinuria occur, and a specific diagnosis of *E. canis* infection is made. The brown dog tick, *Rhipicephalus sanguineus*, transmits *E. canis*; vector competency of this tick in the transmission of *A. platys* is still to be confirmed [6,7]. In experimental infections, *A. platys* infection was found to be self-limiting without treatment after a prolonged period of infection, but *E. canis* infection was controlled only with doxycycline treatment [3]. Co-morbidities with both pathogens are common among dogs in endemic regions and have been reported to alter the pathophysiologic outcome and the severity of the disease [3]. 

Serology based indirect immunofluorescence antibody (IFA) tests and commercially available immunochromatography based point-of-care (POC) tests for *A. platys* and *E. canis* are used in the veterinary field as part of routine annual vector-borne disease screening and as a diagnostic tool when dogs present with clinical signs [8,9]. PCR methods are available for both agents to detect pathogen DNA in the blood and tissues of dogs [10,11,12,13]. *A. platys* has not been cultured in vitro to date. *E. canis* has been cultured in canine macrophage cell lines and tick cell lines, but this procedure is not used for routine agent detection [14]. Both pathogens are obligate intracellular bacteria, and routine bacterial culture, the gold standard for confirmation of bacterial infection is not a practical option for the diagnosis of *A. platys* and *E. canis* diseases. *Anaplasma platys* and *E. canis* invade canine platelets and monocytes, respectively, and replicate in the cytoplasm inside a parasitophorous vacuole forming structure called the “morula.” In *A. platys* infections, morula within the platelet cytoplasm may be seen in stained blood films, but the observation of *E. canis* morula in monocytes is uncommon [3,15,16]. 

In our experience, *A. platys* morula-like structures are occasionally observed in platelets of POC test negative patients with clinical suspicion of *A. platys* infection, and only a proportion of these cases are confirmed as *A. platys* infection by PCR (unpublished data). While false-negative results can be detrimental for patient management, false-positive results may lead to inadvertent treatment using antibiotics. Due to the complex pathogenesis, broad and nonspecific clinical manifestation, and occurrence of co-infection with both agents potentially exacerbating the severity of the disease, diagnosis of *A. platys* and *E. canis* associated diseases can be challenging. In this study, we updated the prevalence of *A. platys* and *E. canis* infection in Saint Kitts, and we assessed the usefulness of PCR and a widely used commercial POC test for diagnosing infections caused by *A. platys* and *E. canis*. Clinical signs induced with *A. platys* and *E.canis* infections can be broad, nonspecific and overlapping and hence evaluation of various clinical pathology parameters tested in a natural setting routinely may help in the diagnostic evaluations [3,16]. Therefore we also evaluated relevant clinical pathology parameters that could potentially be associated with *A. platys* and *E. canis* infection in a proportion of the dogs tested.

## 2. Results

A total of 227 canine blood samples were tested by PCR. Out of these, 188 samples also were tested using the POC test (IDEXX SNAP^®^ 4Dx). According to the manufacturer, this POC test screens for antibodies to *Borrelia burgdorferi*, *A. phagocytophilum,* and *A. platys,* and *E. canis* and *E. ewingii*, and *Dirofilaria immitis* antigen. Since the tick vectors for *A. phagocytophilum* and *E. ewingii* are not present on the island, all *Anaplasma and Ehrlichia* positive cases were considered as *A. platys* and *E. canis* positive in this study [17,18,19]. Twenty seven percent (62/227) of the samples were positive for *A. platys* and 37% (84/227) were positive for *E. canis* DNA by PCR. Co-infection was detected in 43/227 (19%) of the dogs. The POC test was positive in 28% (53/188) of the samples for *A. platys* and 60% (112/188) of the samples for *E. canis.* Among these, 42/188 (22%) canine samples were positive for both agents. Summary of prevalence is shown in Table 1. 

A proportion (65%) of all *A. platys* PCR positive samples were POC test negative, and 26% of the *A. platys* PCR negative samples were POC test positive (Figure 1). In contrast, only 20% of the *E. canis* PCR positive samples were negative by the POC test, but 46% of the *E. canis* PCR negative were POC test positive (Figure 2). We obtained the results of blood film evaluation of 139 canine blood samples out of the 227 samples included in this study. Out of these *A. platys* morula like structures were observed in the platelets of 13 blood films. Of these 13 morula positive samples eight were *A. platys* PCR positive, and five were PCR negative. There was only a fair agreement between the presence of morula and being *A. platys* PCR positive (Figure 3). 

We also compared clinical pathology parameters of 139 of the dogs; data were retrieved from the Ross University School of Veterinary Medicine, Veterinary Diagnostic Laboratory (RUSVM DLAB) database. Anemia was a significant finding among dogs when evaluated using various clinical pathology parameters as suggested by statistically significant associations with low packed cell volume low red blood cell counts, low hemoglobin concentrations, and low hematocrit values. Table 2 shows the clinical pathology parameters that had a significant association with positive PCR or POC test results for both pathogens. Contigency Tables showing all the comparisons of clinical pathology parameters with *A. platys* and *E. canis* test results and *p*-values are included in the Appendix A and reference values in the Appendix A.

## 3. Discussion

The infections caused by *A. platys* and *E. canis* are endemic in the tropical and subtropical geographic regions where the tick vector, *R. sanguineus* population is expanding. Both infections can cause debilitating and life-threatening diseases in dogs. Both *A. platys* and *E. canis* are known as exclusive canine pathogens; however, there are reports suggesting potential zoonotic transmission to humans from Venezuela and the USA [20,21]. In this study, we updated the prevalence of *A. platys* and *E. canis* infections in dogs in Saint Kitts. A study conducted by Kelly et al., in Saint Kitts during two years from 2009–2011 reported serological or PCR evidence of infections with *E. canis* (27%) and *A. platys* (11%) in dogs [22]. Another case-control study during the same period utilized samples from 55 cases of clinically suspected tick-borne diseases and 110 presumably healthy animals [17]. This study reported *A. platys* infection in 0% of clinically suspected cases and 4% of the healthy dogs tested and *E. canis* infection in 35% of the clinical cases and 7% of the healthy group. We recorded a higher prevalence for both pathogens by PCR and the POC tests compared to the above mentioned previous studies conducted on the island. While we could speculate that this could be due to an actual increase in pathogen or vector prevalence, other factors such as the methodology used and population tested may have an effect. Our evaluation of both tests emphasizes the need for careful choice of diagnostic tests and appropriate interpretation of test results to manage *A. platys* and *E. canis* associated diseases in dogs.

Our results identified a difference in agreement between the POC test and PCR results for both agents suggesting the potential for false-positive and false-negative test results when a single test is used. Antibody-based POC tests indicate evidence of exposure over the true presence of infection, and they are widely used in the veterinary field for annual vector borne infection screening and to diagnose diseases caused by *A. platys* and *E. canis* infection in dogs. A number of commercial POC tests are available with varying levels of sensitivity and specificity [8,23]. However, it is important to note the reference test used in many reported test comparisons is not the gold standard test, and defining a gold standard test for these pathogens is difficult to achieve. In contrast to POC tests, PCR detects the presence of *A. platys* and *E. canis* DNA in the infected host and is available only at some diagnostic or reference laboratories. There are few guidelines on using criteria for the diagnosis of *A. platys* and *E. canis* infection in dogs emphasizing the need for careful interpretation of diagnostic tests for patient management [8,24]. Our research results, and clinical experience working in the endemic area studied, reemphasize the use of the combination of both tests to improve the accuracy of *A. platys* and *E. canis* screening and associated disease diagnosis. 

The commercially available POC test used in this study is cost-effective and widely used in veterinary clinics, and according to the test manufacturer, it detects antibody response to two *Ehrlichia* species (*E. ewingii*, *E. canis*) and two *Anaplasma* species (*A. platys* and *A. phagocytophilum*) as well as *Borrelia burgdorferi* antibodies and *Dirofiliaria immitis* antigen. It is reported to be more sensitive than IFA or another commercially available POC test [8,23]. Our data suggest that caution should be exercised when interpreting negative *A. platys* POC test results in dogs with compatible clinical signs, since a high proportion of *A. platys* PCR positive samples were POC test negative. Experimental studies have shown that antibodies to *A. platys* and *E. canis* appear between days 10–24 and 17–35 post-infection, respectively [3]. Hence, as with other antibody tests, there can be a delay between infection and antibody detection. Our study cautions that many of the acute *A. platys* infections may go undetected if serology is used alone; in the case of *E. canis* infection, if PCR alone is used, chronic infection with low parasitemia might be missed. 

Blood film evaluation and the observation of morulae like structures in platelets and monocytes is valuable to presumptively diagnose *A. platys* and *E. canis*, however, the presence of these structures warrants further confirmation using other definitive tests such as PCR. Confirmation is necessary to avoid false-positive results due to the presence of structures arising from products of platelet activation and nuclear remnants of megakaryocytes [13,25,26]. In our study, only a small proportion of the samples tested showed structures resembling morula, and only a proportion (61%) of those were confirmed as *A. platys* PCR positive and hence false-positive results using this method need to be emphasized. The trend among veterinary practitioners is to use either POC or PCR to diagnose *A. platys* and *E. canis* infection. In suspected clinical cases or in endemic areas, performing serology and PCR tests concurrently is useful to detect early infection and to avoid false-negative PCR results due to potential low bacteremia with both pathogens that may handicap target DNA amplification. 

There is a marked difference in pathogenesis and clinical signs of *E. canis* and *A. platys* infection in dogs. We found significant associations between positive *A. platys* or *E. canis* test results and changes in multiple clinical pathology parameters suggestive of anemia, leukocytosis, lymphcytosis, hyperprotenemia, hypoproteinemia, and eosinophilia in a proportion of the dogs tested. *Anaplasma. platys* infection leads to platelet infection and acute symptoms associated with thrombocytopenia, whereas *E. canis* causes more debilitating disease due to chronic infection, and co-infection may exacerbate clinical outcomes. Lymphocytosis in dogs with canine ehrlichiosis has been previously described and is presumably the result of antigenic stimulation by the organism [27]. The monocytosis may also be a component of an inflammatory response, Monocytosis was observed in 46% of dogs infected with *E. ewingii,* but monocytosis is not a routine observation in *E.canis* infection [27]. A significant proportion of dogs in this study had eosinophilia, and the incidence of eosinophilia observed in these dogs is most likely a result of concurrent endo and ectoparasitic infections, including babesiosis, and hepatozonosis are common in island dogs [17,22]. Thrombocytopenia, one of the consistent hematologic abnormalities in dogs infected with *E. canis,* has been attributed to vasculitis, immune-mediated destruction, platelet destruction, and sequestration, and suppression of platelet production [28]. Thrombocytopenia is also common in dogs infected with *A. platys* and is thought to be a consequence of platelet phagocytosis by macrophages, either because of injury to platelets by the organism or immune-mediated destruction [29]. Platelets are recently known to play an active role in innate immune mechanisms [30]. Therefore, *A. platys* infection, even asymptomatic, may dampen the immune response in infected dogs and may increase the susceptibility to other infections. Further well designed investigations in a naturally exposed canine population in an endemic area such as Saint Kitts may unravel the factors related to transmission, infection dynamics and pathogenesis of *A. platys* and *E. canis* infections in dogs and potential underlying mechanisms that might cause these type of clinicopathological abnormalities. We did not compare the test results to symptomatic and asymptomatic status in dogs due to the nature of the population tested. However, infections with both pathogens can have broad manifestaions and outcomes from asymptomatic subclinical infections to severe life-threatening or chronic debilitating disease. Subclinical infection often remains undetected and periodic screening and subsequent treatment of infected dogs in endemic regions may reduce the reservoir status, transmission and the progression to severe or chronic disease.

With the advances in the genomic technology, sequencing of the *E. canis* whole genome was made possible due to its ability to grow in cell lines [31]. In contrast, *A. platys* has never been grown in culture and the whole genome sequencing of *A. platys* was made possible only recently from our laboratory, by applying a metagenomic approach using a blood sample collected from an acutely infected canine patient from this study area [32]. In addition to advancing the knowledge on the evolutionary and genomic characteristics of these complex pathogens, availability of the genome data will improve our understanding of the pathogenesis, facilitate the development of improved diagnostics, and may pave the way to protective vaccines.

## 4. Materials and Methods 

The study was performed from May 2017 to May 2018 on Saint Kitts, West Indies, a small Caribbean island located in the Lesser Antilles. We used a convenient sampling strategy to collect blood samples from canine subjects with all collection procedures adhering to the RUSVM approved institutional animal care and use (IACUC) protocol (Protocol # 17.04.21). Random blood samples collected from client-owned dogs presented to Ross University Veterinary Clinic (RUVC), blood samples submitted to Ross University School of Veterinary Medicine, Veterinary Diagnostic Laboratory (RUSVM DLAB), and from local households were used in this study. Venous blood was collected into tubes containing potassium ethylenediaminetetraacetic acid (EDTA) as per standard practice at the RUVC and as per the study protocol. 

After blood collection, the POC test (IDEXX SNAP^®^ 4Dx, IDEXX Laboratories, Westbrook, Maine) was performed either as part of the clinical assessment of the dog or specifically for this study. For the POC test used in this study, is reported to have high sensitivity and specificity for the detection of *Anaplasma* and *Ehrlichia* in comparison to reference tests used [33]. Blood was either immediately used for DNA extraction or stored at −80 °C until further processing. Since samples were convenience samples and included leftover samples from the RUSVM DLAB, POC was not performed in all of the cases. Blood samples were tested for the presence of *A. platys* and *E. canis* DNA using a semi-quantitative real-time PCR method described previously [12,13]. DNA from 200 μL of blood was extracted using the High Pure PCR Template Preparation Kit (Roche Life Sciences, Indianapolis, Indiana) following the manufacture’s protocol. Extracted DNA was used for PCR on the same day or stored at −80 °C until PCR analysis. Canine blood samples previously confirmed to be positive for both *A. platys* and *E. canis* were used as the positive control and sterile water as the negative control for PCR. A cycle threshold value of 40 and above was considered negative. We also retrieved complete blood count (CBC) results performed using the Abaxis Vetscan HM5™ electronic cell counter (Abaxis, Union City, CA, USA) and the results from manual blood film evaluation when available. These procedures were performed as part of routine analyses of dogs seen at the RUVC. The data were compared with the POC and PCR test results. 

Data Analysis: Data were analyzed using GraphPadPrism (version 8.3.0)(GraphPad Software, La Jolla, CA, USA). To assess agreement between tests, Cohen kappa was calculated using GraphPad Software QUICK Calc. The results were interpreted using the classification of Landis and Koch [34]. We compared *A. platys* and *E.canis* PCR and POC test results to each of the clinical pathology parameters using Fisher’s Exact test. Statistical significance was set at *p*-value ≤0.05.

## 5. Conclusions

In conclusion, in suspected cases of *A. platys* and *E. canis* infection, the careful choice of diagnostic tests is crucial for the accurate diagnosis of the various stages of infection, and subsequent patient management. We suggest the use of a combination of serology and PCR tests along with a thorough evaluation of clinical parameters to improve the confidence in the diagnosis and management of *A. platys* and *E. canis* infection in dogs. While PCR could be beneficial to confirm the early stage of disease when antibody levels are low or undetectable, the POC test may be helpful in animals with low or undetectable bacteremia.

## Figures and Tables

**Figure 1 pathogens-09-00488-f001:**
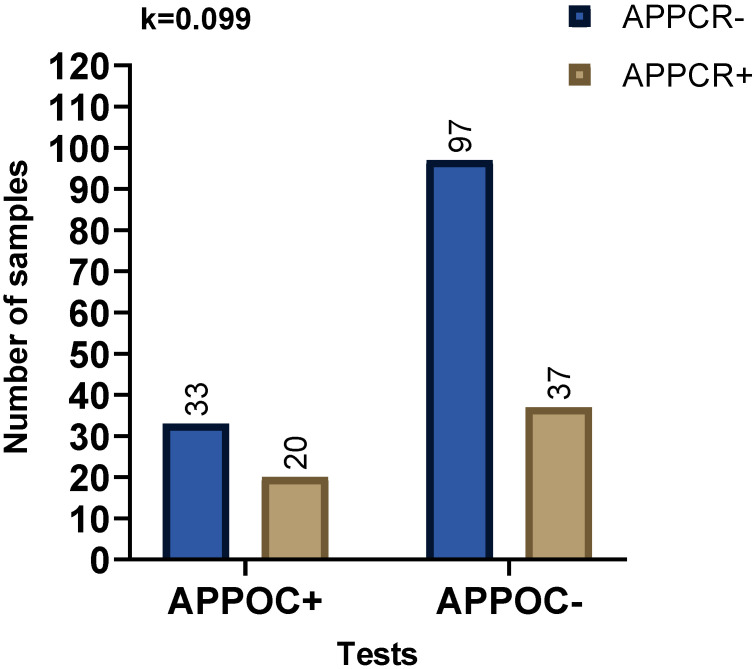
Comparison of agreement between the POC test results with PCR for the diagnosis of *A. platys* (AP) infection. Fisher’s exact test was used for comparisons, and Cohen’s Kappa value was calculated. Observed Kappa (k) value shown on the graph is interpreted, as agreement between tests is “slight.” APPCR: *A. platys* PCR; APPOC: *A. platys* POC result.

**Figure 2 pathogens-09-00488-f002:**
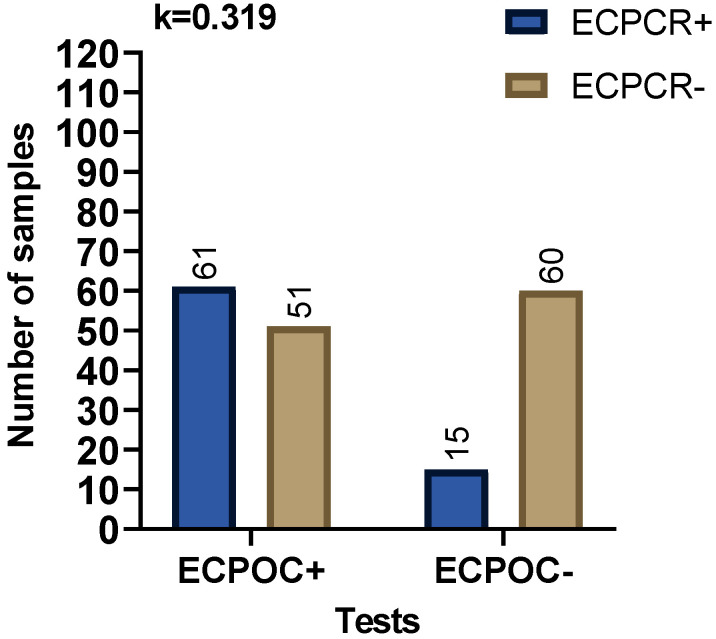
Comparison of agreement between POC test results and PCR for the diagnosis of *E. canis* (EC) infection. Fisher’s exact test was used for comparisons, and Cohen’s Kappa value was calculated. Observed Kappa (k) value shown on the graph is interpreted, as agreement between tests is “fair.”. ECPCR: *E. canis* PCR; ECPOC: *E. canis* POC.

**Figure 3 pathogens-09-00488-f003:**
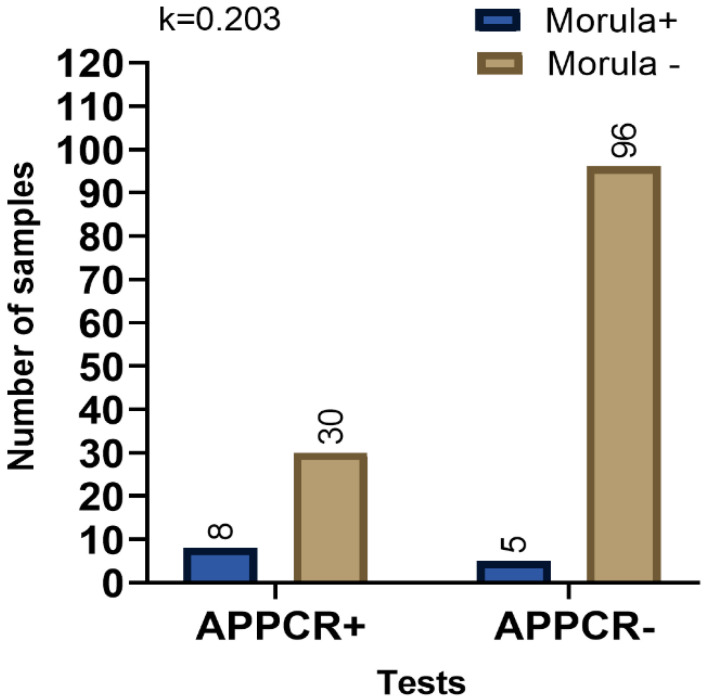
Comparison of agreement between the observation of platelet morula and *A. platys* (AP) PCR. Fisher’s exact test was used for comparisons, and Cohen’s Kappa value was calculated. The observed Kappa (k) value shown on the graph is interpreted, as agreement between tests is “fair.” APPCR: *A. platys* PCR.

**Table 1 pathogens-09-00488-t001:** *A. platys* and *E. canis* prevalence in dogs in Saint Kitts using PCR and a commercial point-of-care (POC) test, May 2017–May 2018.

Tests	Number of Positive/Number Tested (Percentage: 95% Confidence Interval)		
	*A. platys*	*E. canis*	Both Agents
POC Test	53/188 (28%; 22–35)	112/188 (60%; 53–67)	42/188 (22%; 17–29)
PCR	62/227 (27%; 22–33)	84/227 (37%; 31–44)	43/227 (19%; 15–25)

**Table 2 pathogens-09-00488-t002:** Clinical pathology parameters observed in dogs with significant association with a positive test result. Each of the clinical pathology parameters were compared with test results using Fisher’s exact test, and *p*-values less than 0.05 were considered as significant. Appendix A contains individual contingency tables (Appendix A) and reference values (Appendix A).

Clinical Pathology Parameter	Positive Test Results with Significant Associations (*p*-Value)
Low packed cell volume	AP-PCR (*p* = 0.0244) EC-PCR (*p* = 0.0007) EC-POC (*p* = 0.0077) AP and EC PCR (*p* = 0.0011)
Low Red blood cell count	EC-PCR (*p* = 0.0013) EC-POC (*p* = 0.0479) AP and EC PCR (*p* = 0.0249)
Low hemoglobin concentration	AP-PCR (*p* = 0.0009) EC-PCR (*p* = 0.0001) EC-POC (*p* = 0.01) AP and EC-PCR (*p* = 0.0001)
Low hematocrit values	AP-PCR (*p* = 0.0051) EC-PCR (*p* = 0.0001) EC-POC (*p* = 0.0050) AP and EC-PCR (*p* = 0.0001)
Low plasma protein concentration	AP-POC (*p* = 0.0359)
High plasma protein concentration	AP-PCR (*p* = 0.0002) EC-POC (*p* = 0.0001) AP and EC-PCR (*p* = 0.0041) AP and EC-POC (*p* = 0.0023)
Lymphocytosis	AP-PCR (*p* = 0.0466) EC-PCR (*p* = 0.0003) AP and EC-PCR (*p* = 0.0055)
Monocytosis	EC-PCR (*p* = 0.0199)
Eosinophilia	AP-PCR (*p* = 0.0090) EC-PCR (*p* = 0.0332) AP and EC-PCR (*p* = 0.0189)
Thrombocytopenia	AP-PCR (*p* = 0.0199) EC-PCR (*p* = 0.0001) AP-POC (*p* = 0.001) EC-POC (*p* = 0.0088) AP and EC-PCR (*p* = 0.0002) AP and EC-POC (*p* = 0.0001)

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
