# Peer review of "Serologic and Molecular Diagnosis of Anaplasma platys and Ehrlichia canis Infection in Dogs in an Endemic Region"

_pathogens, 2020, doi:10.3390/pathogens9060488_

Round 1

Reviewer 1 Report

This study examines the use of a point-of-care serology test and PCR for the diagnosis of Anaplasma platys and Ehrlichia canis infected dogs inhabiting the Caribbean island of Saint Kitts. It is concluded that the choice of diagnostic tests is essential for the accurate diagnosis of the various stages of infection and subsequent veterinary care. It is suggested that the use of a combination of serology and PCR tests, along with an evaluation of clinical parameters, is required to improve the diagnosis and management of A.platys and E. canis infection in dogs.

The paper is well written and appropriately referenced. The materials and methods section could be improved, for example, it isn't clear how the 227 cases (PCR) were selected vs the 187 cases (POC serology). Were these samples also submitted for follow up testing using a 'gold' standard technique ? Do we know the DSe and DSp of the serology test used ? Were the 187 serology samples taken from the same dogs as the PCR samples screened ? How was the sampling plan deigned ? Do we know the age groups or gender of the dogs tested ? Had any of these animals been treated for the condition diagnosed ?

The Figures could include more explanation (in the legend) of the abbreviations used. Figure 3 has referred to 45 samples results vs n=46 in the text. The tables in the supplementary data are also hard to follow and would be improved if there was more explanation in the Table legends.

Author Response

Reviewer 1

This study examines the use of a point-of-care serology test and PCR for the diagnosis of Anaplasma platys and Ehrlichia canis infected dogs inhabiting the Caribbean island of Saint Kitts. It is concluded that the choice of diagnostic tests is essential for the accurate diagnosis of the various stages of infection and subsequent veterinary care. It is suggested that the use of a combination of serology and PCR tests, along with an evaluation of clinical parameters, is required to improve the diagnosis and management of A.platys and E. canis infection in dogs.

We would like to thank the reviewer for the time and effort taken to review this manuscript and very useful comments

The paper is well written and appropriately referenced. The materials and methods section could be improved, for example, it isn't clear how the 227 cases (PCR) were selected vs the 187 cases (POC serology).

There is a small correction from 187 to 188, the number of samples tested by POC. Our apologies for the error.

Out of 227 samples, the POC test performed with 188 samples. This is due to the use of left over clinical samples within the study and not all clinical samples were tested via POC at the discretion of the examining veterinarian. We have clarified this in the materials and methods. All the samples tested by POC were tested by PCR. However, a proportion of samples tested by PCR was not tested by POC.

Were these samples also submitted for follow up testing using a 'gold' standard technique ? Do we know the DSe and DSp of the serology test used ? Were the 187 serology samples taken from the same dogs as the PCR samples screened ?

Unfortunately, considering the nature of these pathogens, appropriate gold standards are not well defined. We have expanded the discussion to explain this issue.

Yes, all samples tesed with the POC were tesed with PCR.

How was the sampling plan deigned ? Do we know the age groups or gender of the dogs tested ? Had any of these animals been treated for the condition diagnosed?

This was a convenience sample design. We have the gender and age information from samples collected through RUVC but not from the field. Since the study was performed to assess the prevalence and compare the tests, we did not want to compare any confounding variables, hence not included. We do not have any data on treatment. The dogs coming through the clinic might be treated after obtaining test results, however, it is very unlikely that field animals are treated.

The Figures could include more explanation (in the legend) of the abbreviations used.

We have expanded the figure legends.

Figure 3 has referred to 45 samples results vs n=46 in the text.

 The tables in the supplementary data are also hard to follow and would be improved if there was more explanation in the Table legends.

Thanks for noticing the mistake. However, we found that we have information on more animals on CBC data and decided to reanalyze the data with those numbers and present the corrected numbers in this revised version.

We have improved the data presentation in the supplementary table.  We have included a new table in the main body of the manuscript. In addition, In the revised version, in the supplementary file 1, we present contingenecy tables showing  comparison of  each of clinical pathology parameter  with test results. We are sure that this change has improved the clarity of this data.

Reviewer 2 Report

The manuscript by Lara and colleagues evaluates the use of PCR and a commercial point of care immunoassay test for detecting two intracellular pathogens (A. platys and E. canis) in blood from dogs in St Kitts. The results are straightforward, but highlight potential issues with each test. The manuscript is well written although a few suggestions for improvement are provided below. Please note that point 1 is not a discretionary edit.

  1. L21. The three calculations seem incorrect; they should be 28%, 60% and 22%.
  2. The authors use two styles of spacing before references; +/- a space. I recommend that spaces be inserted prior to refs on L36, L40, L42, L44, L51, L125, L128, L145, L148, L156, L168, L170, L175, L178, L179, L205.
  3. References should be included for statements of facts L49, L55, L56, L57, L74, L173
  4. L53 delete the word “used”
  5. L67 be associated with
  6. Unless it is against journal style, it would be useful to identify the POC test and the source of the canine patients earlier in the manuscript. This information only comes to light after reading most of the manuscript (due to the placement of the Methods section in this journal).
  7. In Table 1, only one of the 6 percentages has a decimal place. They should all have this degree of accuracy or none. Also, there is a space before the % sign for 1 of 6. This should be unified.
  8. L135 delete the second use of “infection” in this line
  9. L154 “however” does not require italicization
  10. L168 abbreviate “Ehrlichia” to E. Also it should be E. canis on L170 (please add period)
  11. L183. Should “a” before antibody be “and” or “an”? It requires modification.
  12. L198 Use “California” in full to match style for other states listed.
  13. L202 La Jolla,
  14. L207 value (spelling)
  15. L218 A. platys (spelling and spacing issues)
  16. L241 red font needed?
  17. L264 Does “Borne” need a period, since not an abbreviation?
  18. L128. Please include in the text the previous estimates of incidence, so that the reader can put your data into context, without having to look up the data in the two references cited.
  19. Methods. What were used as positive and negative controls for each assay and for each organism. Were any of the discrepant results re-tested?

Author Response

Reviewer 2

The manuscript by Lara and colleagues evaluates the use of PCR and a commercial point of care immunoassay test for detecting two intracellular pathogens (A. platys and E. canis) in blood from dogs in St Kitts. The results are straightforward, but highlight potential issues with each test. The manuscript is well written although a few suggestions for improvement are provided below. Please note that point 1 is not a discretionary edit.

Thank you very much for the thorough review and valuable suggestions. We have made all the requested corrections.

  1. L21. The three calculations seem incorrect; they should be 28%, 60% and 22%.

Corrected, Thank you for noticing this significant error. Our apologies for making this error.

  1. The authors use two styles of spacing before references; +/- a space. I recommend that spaces be inserted prior to refs on L36, L40, L42, L44, L51, L125, L128, L145, L148, L156, L168, L170, L175, L178, L179, L205.

Thanks for pointing out this. We used Mendeley software to insert references and now noted these discrepancies.  Corrected to suit the Journal format.

  1. References should be included for statements of facts L49, L55, L56, L57, L74, L173

Included

  1. L53 delete the word "used"

Deleted

  1. L67 be associated with

Corrected

  1. Unless it is against journal style, it would be useful to identify the POC test and the source of the canine patients earlier in the manuscript. This information only comes to light after reading most of the manuscript (due to the placement of the Methods section in this journal).

Journal style is to place the materials and methods at the end. However. In the revised manuscript, we mentioned the name of the POC test used in the results section.

  1. In Table 1, only one of the 6 percentages has a decimal place. They should all have this degree of accuracy or none. Also, there is a space before the % sign for 1 of 6. This should be unified.

Corrected

  1. L135 delete the second use of "infection" in this line

Deleted

  1. L154 "however" does not require italicization

Changed

  1. L168 abbreviate "Ehrlichia" to E. Also it should be E. canis on L170 (please add period)

Changed

  1. L183. Should "a" before antibody be "and" or "an"? It requires modification.

Modified

  1. L198 Use "California" in full to match style for other states listed.

Changed

  1. L202 La Jolla,

Changed

  1. L207 value (spelling)

Corrected

  1. L218 A. platys (spelling and spacing issues)

Corrected

  1. L241 red font needed?

Corrected

  1. L264 Does "Borne" need a period, since not an abbreviation?

Corrected

  1. L128. Please include in the text the previous estimates of incidence, so that the reader can put your data into context, without having to look up the data in the two references cited.

Included- L139-L145

  1. Methods. What were used as positive and negative controls for each assay and for each organism. Were any of the discrepant results retested?
  2. canis and A. platys positive canine blood samples were used for  PCR.

Our intention was to compare these tests to evaluate this discrepancy as both tests are independently used by veterinarians to screen and diagnose diseases associated with these agents. Initially, we retested some of the samples but did not continue it to save time and resources.